# Development of the Microemulsion Electrokinetic Capillary Chromatography Method for the Analysis of Disperse Dyes Extracted from Polyester Fibers

**DOI:** 10.3390/molecules27206974

**Published:** 2022-10-17

**Authors:** Anna Sałdan, Małgorzata Król, Daria Śmigiel-Kamińska, Michał Woźniakiewicz, Paweł Kościelniak

**Affiliations:** 1Laboratory for Forensic Chemistry, Department of Analytical Chemistry, Faculty of Chemistry, Jagiellonian University in Krakow, Gronostajowa 2, 30-387 Kraków, Poland; 2Laboratory of Chemical Analytics and Diagnostics, Department of Environmental Analytics, Faculty of Chemistry, University of Gdańsk, Wita Stwosza 63, 80-308 Gdansk, Poland

**Keywords:** disperse dyes, microtraces, polyester fibers, microemulsion electrokinetic capillary chromatography, MEEKC, electrophoresis capillary, forensic examination

## Abstract

The study aimed to develop a method for the separation of dispersed dyes extracted from polyester fibers. Nine commercially available disperse dyes, which were used to dye three polyester fabrics, were tested. Extraction of dyes from 1 cm long threads was carried out in chlorobenzene at 100 °C for 6 h. The separation was performed using microemulsion electrokinetic capillary chromatography (MEEKC) with photodiode array detection. Microemulsion based on a borate buffer with an organic phase of n-octane and butanol and a mixture of surfactants, sodium dodecyl sulphate and sodium cholate, were used. The addition of isopropanol and cyclodextrins to microemulsion resulted in a notable improvement in resolution and selectivity. The content of additives was optimized by using the Doehlert experimental design. Values of the coefficient of variance obtained in the validation process, illustrating the repeatability and intermediate precision of the migration times fit in the range of 0.11–1.24% and 0.58–3.21%, respectively. The developed method was also successfully applied to the differentiation of 28 real samples—polyester threads collected from clothing. The obtained results confirmed that proposed method may be used in the discriminant analysis of polyesters dying by disperse dyes and is promisingly employable in forensic practice.

## 1. Introduction

Fibers are an interesting research material, especially in forensic laboratories. They are commonly found at the crime scene and are often used as evidence in court cases, as they can confirm the presence of a person at a given location or the contact between two surfaces due to the transfer of microtraces.

Nowadays, clothing is more and more often made of synthetic fabrics. Polyesters are very resistant to mechanical, physical, and biological damage. They are also resistant to stretching, crushing, sunlight, acids and alkalis, and they poorly absorb water. Polyester fabrics do not have thermal insulation properties. In 2019, polyester fibers represented 52.2% and cotton only 23.2% of global fiber production (~111 million metric tons) [1,2]. Such a common occurrence of polyester fibers causes a problem in forensic analyses because they lose their evidential value.

For the analysis of fiber microtraces, nondestructive methods, mainly spectroscopic [3,4,5,6,7,8,9,10,11,12], are used initially. However, sometimes they are not able to give an unambiguous answer about the similarity of the evidence sample and comparative material if the sample does not have individual features. Therefore, methods are sought that will be able to distinguish fibers, mainly due to the chemical composition of the dyes used. Numerous varied chromatographic methods from thin-layer chromatography (TLC) and high-performance thin-layer chromatography (HPTLC) [13] to HPLC [14,15,16,17,18] to capillary electromigration methods (CEMs) [19,20,21,22,23,24,25,26,27,28,29,30,31,32] with different detections have been reported so far. Although capillary electrophoresis is destructive as it requires the extraction of dyes, it seems to be a highly promising method which can significantly increase the degree of discrimination against similar fibers. Benefits and problems related to the use of various CEMs for different dye groups have been extensively and discussed in-depth in the review [33].

The subject of interest of these studies is a group of disperse dyes used mainly for dyeing man-made fabrics such as polyester. They are non-ionic water-insoluble or poorly soluble dyes. This name originates from the mechanism of dye application to the fibers. In terms of their chemical structure, disperse dyes do not constitute a uniform group; it includes azo, anthraquinone, benzodifuranone, nitro, and other dyes. The dyeing process requires the usage of a suitable dispersing agent in an acidic medium in the form of suspensions. In such conditions, the disperse dyes are applied to the fibers by organic carriers, at high temperature or pressure. The dye particles diffuse into the fibers by adsorption, are retained by physical forces, weak van der Waals forces, or hydrogen bonds [34]. The commercially available dyeing mixture, in addition to dyes, also contains numerous additives that enable or improve the process of textile dyeing. The most likely additives are dispersants that facilitate the appropriate dispersion of the dye molecules in the dyeing bath. During the dyeing process, only single dye particles or their dimers can penetrate the interior of polyester fibers, therefore the presence of dispersants (e.g., lignin sulfonates, naphthalene sulfonate, formaldehyde condensates) is often necessary. Other possible additives are carriers that facilitate the diffusion of dye molecules into the dyed fiber or the surface-active agents that enable solubilization of the dyes, and thus accelerate the dyeing process [2,35,36].

In the authors’ opinion, a promising prospect for the analysis of complex samples such as disperse dyeing mixtures is the application of microemulsion electrokinetic capillary chromatography (MEEKC). This is CEM in which microemulsion (ME) is used as the background electrolyte (BGE). It is worth recalling here that ME consisting of four main components (water phase, oil phase, surfactant, and co-surfactant) mixed in appropriate proportions is a thermodynamically stable dispersion system, which enables the analysis of water-insoluble or poorly soluble substances. Moreover, not only electrophoretic and chromatographic interactions occur during separation, but attraction or repulsion between microdroplets and analytes can also improve the process [37]. MEEKC has aroused growing interest among researchers since 1991 when the first publication on this technique was published [38]. Many articles in the literature report a wide range of applicability of this technique [37,39,40,41,42,43]. Reviews clearly show that MEEKC can be used for various compounds and that its unique properties are especially useful for complex mixtures containing analytes with varying charge and hydrophobicity. However, there are no scientific reports on the use of MEEKC to analyze textile fibers and their dedicated dyes.

Another important analytical challenge is the development of an efficient extraction procedure compatible with MEEKC, as some organic solvents or their elevated concentration may affect the stability of ME. It is worth noting here that there have also been published reports concerning the type of solvent used to extract disperse dyes from polyester fibers. Among them, methanol, acetonitrile, or dimethyl sulfoxide can be found. However, by far the most common is the extraction with chlorobenzene at a temperature elevated to 100 °C or higher [44,45]. A recent study using LC-MS also indicates the efficiency of chlorobenzene as a solvent, both in terms of the number of extracted substances and the yield [46]. It is worth noting that these reagents were not used for MEEKC, however.

Therefore, the main goal of this research was to develop an innovative method of extraction and separation of disperse dyes extracted from polyester fibers with the use of microemulsion electrokinetic capillary chromatography for forensic purposes. 

## 2. Results and Discussion

### 2.1. Preliminary Tests of the BGE Components

The most common electromigration capillary method applied to the analysis of neutral compounds (examples of which are disperse dyes) is micellar electrokinetic capillary chromatography (MEKC). Therefore, one of the variants of MEKC, non-aqueous micellar electrokinetic capillary chromatography (NAMEKC), was tested first. To ensure good solubility of the analytes, BGE was prepared based on an organic solvent—DMSO. The exact composition of BGE1 is presented in Appendix A. In addition, the use of DMSO in both BGE and extractant would shorten the sample preparation process (the sample may be analyzed immediately after the extraction step). However, the use of NAMEKC did not guarantee satisfactory separation efficiency. Moreover, the relatively high cut-off value of DMSO (268 nm) decreases analytical sensitivity, since most analytes have an absorption band at ca. 220 nm. Then, the developed method would be able to detect mainly the presence of dyes but not the colorless substances, which may also be important information in the process of sample differentiation.

In the next stage, MEEKC was used in a variant with microemulsions composed of water. Due to the presence of the hydrophobic oil phase, they ensured good solubility of the dyes. A series of MEs was tested (ME1–ME8.4 composition is summarized in Appendix A). During the preliminary tests, the following criteria were assessed: resolution, level of current, and time of analysis. The best separation efficiency was obtained after using ME8.1, in which stock ME8 (consisting of 0.8% n-octane as the oil phase, 6% butanol as a co-surfactant, a mixture of two anionic surfactants: 1.5% SDS and 1.5% sodium cholate, and 90.2% of the water phase: 40 mM sodium borate) was mixed with isopropanol (9:1, *v*/*v*, see Appendix A). Such an approach gave the best results in terms of separation efficiency but resulted in a long migration time of analytes (the total analysis time was 60 min, whereas, e.g., ME4 40 min, and for BGE1 30 min), which may lead to deterioration in reproducibility of the method. However, the adverse effect of isopropanol on migration times can be compensated by increasing the separation temperature. Therefore, the ME8.1 microemulsion was chosen for further optimization despite the long analysis time.

Other frequently used BGE modifiers that significantly improve the separation process are cyclodextrins. In the preliminary research, the addition of two cyclodextrins (CD), 2-hydroxypropyl-β-cyclodextrin (HP-β-CD) and 2-hydroxypropyl-γ-cyclodextrin (HP-γ-CD) separately (ME8.2, ME8,3) and their mixture in a 1:1 molar ratio (ME8.4), to ME8.1 was tested at a concentration of 25 mM (composition is shown in Appendix A). These cyclodextrins were chosen due to their good solubility in both water and isopropanol (the main components of ME8.1). The best result concerning all three above-mentioned criteria was obtained using a mixture of HP-β-CD and HP-γ-CD (ME8.4). This is probably due to the difference in ring diameter of both compounds, which affects the interaction with separated analytes. 

More details on the results obtained in the preliminary studies can be found in Appendix A.

### 2.2. Optimization of Separation Conditions by Using Doehlert Experimental Design

For the optimization process, the three-factor Doehlert experimental design combined with the response surface methodology was selected. This methodology allowed the optimization of the isopropanol and cyclodextrin concentrations simultaneously, and the separation temperature. Moreover, it enabled optimization to be performed based on a relatively small number of experiments (13), thus limiting the time and costs of the optimization process. The isopropyl alcohol content was established in the range 0–30% at 7 levels, the total cyclodextrin concentration in the range 20–60 mM at 5 levels, and the separation temperature in the range 25–35 °C at 3 levels.

The experimental Doehlert matrix design with coded and real values of variables is presented in Table 1. The relationship between these variables is expressed by the following equation: (1)$$x_{i}=\left\{\frac{c_{i}-c_{i}^{0}}{∆c_{i}}\right\}\alpha$$
where *x_i_* is the value of the coded variable, *c_i_* is the value of the real variable, *c_i_*^0^ is the center value of the real variable, ∆*c_i_* is the range between the maximum and the minimum value of real variables, and α is the range for a coded variable. 

The separation time was 30 min. As a system response, the response function *F* was constructed:(2)$$F=\frac{n}{t_{D1}}a$$
where *n* is the number of dye signals (dye peaks, indicated based on absorption bands in the Vis spectrum), and *t_D_*_1_ is the migration time of the red dye (D1) present in the separated mixture of nine dyes standards (D1–D9, details in Section 3.2 in Table 6). The migration time of D1 was selected after careful manual inspection of the experimental results collected. The first and most important reason was the confirmed presence of a peak corresponding to D1 on all 13 electropherograms obtained under different experimental conditions. Second, it was also the most intense signal in the Vis range (providing high-quality Vis spectra, used for dye identification). Parameter *a* was introduced to reduce the influence of the experimental points that resulted in the improper shape of the baseline of electropherograms (highly elevated during measurements). It was set to 1, for experimental points where a flat baseline was observed, or 0 for experimental points where the baseline was elevated. Such defined response function *F* allowed for the selection of conditions ensuring the best resolution (measured by *n*) in the shortest possible time (in relation to *t_D_*_1_). The experimental plan with the *F* values determined during the experiments is presented in Table 1.

In the next step, the surface response methodology was employed to estimate the optimal separation conditions. Based on the values of the *F* function, a quadric polynomial response model was determined:(3)$$y=\beta_{0}+\overset{n}{\underset{i=1}{\sum }}\beta_{i}x_{i}+\overset{n}{\underset{i=1,j>i}{\sum }}\beta_{ij}x_{i}x_{j}+\overset{n}{\underset{i=1}{\sum }}\beta_{ii}x_{i}^{2}$$
where *y* is the system response, and *β*_0_, *β_i_*, *β_ii_*, and *β_ij_* (intercept, linear, interaction, and quadric, respectively) are the coefficients determined using linear regression. The statistical analysis of the significance of the coefficients obtained (details in Appendix A), shows that the separation process is influenced by factors: c_iPr_, c_iPr_^2^, c_iPr·T_. It is worth emphasizing that the amount of cyclodextrins in the tested range of concentrations does not have a significant influence on the separation efficiency. 

The values of the desirability function (*d*) were determined by the Statistica 13.3 PL software. This function assumed 0 for the completely undesirable system response and 1 when the response obtained was the most satisfactory. The graphs showing the response surfaces for a given pair of independent variables obtained (the remaining variable was kept at the optimal level) are presented in Figure 1.

The response surfaces were used to estimate the optimal values of the independent variables. Figure 1a indicates the local maximum for the coded value of the isopropanol concentration equal to 0.886 (corresponding to the value 30%). The amount of cyclodextrins present in the BGE has no significant influence, however, the maximum of the *d* function is located at the coded value 0 (corresponding to the value 40 mM). 

The dependence of the function *d* on T (Figure 1b,c) clearly shows two maximum areas that are outside the study space. Typically, in such cases, the range of variables under study should be extended to find the maximum of the desirability function that reflects the best separation condition. In this case, however, this is undesirable, since a temperature increase above 35 °C can contribute to a change in BGE composition during a long measurement sequence, reducing repeatability. A higher separation temperature reduces Joule heat dissipation, further reducing repeatability. In addition, temperature instability drives to higher variability of separation current affected by the temperature dependence of conductivity. Holding the capillary temperature requires switching on and off Peltier element in the thermostat which induces the waveform of the current oscillating around a certain value (at 35 °C it was around 37.5 μA). With increasing temperature, an increase in the amplitude and frequency of oscillations was observed. There was a risk that when the temperature was higher than 35 °C, the fluctuations would be large enough to negatively affect the separation.

Thus, the optimal values found were: c_iPr_ = 30%, c_CD_ = 40 mM, and T = 35 °C. The electropherogram registered for the dyeing mixtures (D1–D9) under the optimal separation conditions is shown in Figure 2. It was possible to separate the components of the mixture in less than 30 min.

The identification of dyes D1–D9 was performed by comparing measured migration times with those collected during the analysis of single component standards. One should note that the separation of multicomponent mixtures is a complex task and unexpected shifts of analytes may be observed due to (i) the interaction between dyes and/or excipients or other compounds in this complex system; (ii) the competitive nature of equilibrium between analytes and CDs; (iii) physicochemical changes of sample zone, e.g., conductivity or viscosity. These phenomena may affect the selectivity of the method and cause changes in the migration times of analytes. However, this analytical challenge may be circumvented by a collection of the library of UV-Vis spectra of individual dyes, which in our case strongly supported the identification of dispersive dyes.

### 2.3. Optimization of the Extraction Parameters

Several extractants dedicated to dispersing dyes known from the literature (see Section 1), including MeOH, ACN, DMSO, dichloromethane, and chlorobenzene were tested. Two types of extraction were used—high-temperature extraction (HTE) and ultrasound-assisted extraction (UAE). The evaluation of the extraction efficiency was carried out visually, assessing the intensity of the extract hue and the color of the polyester threads after extraction (it was assumed that the whiter the color of the threads, the better the extraction). The best result was obtained for HTE with chlorobenzene at a temperature of 100°C. Therefore, it was decided to use this method in further research. 

The previously developed separation method was used to assess the effectiveness of the extraction conditions. The extraction was carried out from four standard polyester fabrics, dyed with dyeing mixtures of known compositions, FA, FB, FC, and blank sample FD (see Table 6 in Section 3.2). Two lengths of threads, 1 and 4 cm, and different extraction times of 0.5, 1, and 6 h were tested. Since the purpose of the research was to develop a method for qualitative analysis of the components of polyester fiber extracts, the total number of signals (both dye peaks and peaks corresponding to colorless substances) observed on the electropherograms was selected to assess the effectiveness of the extraction conditions (excluding the signals generated by the electroosmotic flow, EOF). The results are shown in Figure 3.

The data in Figure 3 show that the results obtained are divided into two groups. The first one contains extracts obtained from 1 cm long threads for 0.5 and 1 h, respectively. Under the compared conditions, the number of signals for each of the reference fabrics analyzed was the same. The second group includes extracts obtained from threads of 4 cm for 6 and 1 h, respectively, and from threads of 1 cm for 6 h. The obtained results are practically the same for each fabric except for FB under the conditions of 4 cm, 6 h, where one signal more was observed. Comparing the results for both groups, in the second group, the number of signals is much greater. Especially in the case of FA-FC fabrics that have undergone a dyeing process. Interestingly, the blank sample FD also provides signals detected on the electropherograms. The UV-Vis spectrum analysis stands for that they originate from colorless substances, also present in the fibers. The detection of these substances, in addition to information on dyes, can also be an important factor in the differentiation process.

From the point of view of forensic examination of fibers, the main challenge is a small amount of available material. For this reason, particular attention was put on minimizing the dimensions of the extracted sample while at the same time having the largest possible number of signals. Therefore, 6 h of extraction from 1 cm long threads were selected as optimal.

### 2.4. Method Validation

In the next step of these tests, the developed procedure was validated in order to control it and confirm whether the intended objectives regarding the use of the method were met. Taking into account the qualitative character of the developed method the particular impact was put mainly on the repeatability of the method, as this is the most important parameter for qualitative and comparative analyses.

The nine standard dye solutions dissolved in the BGE (for details, see Section 2.5) were analyzed, testing the injection and sampling repeatability as well as the intermediate precision. The examination of the repeatability of the injection consisted of performing four repetitions of one sample, whereas the repeatability of the sampling was established according to the results obtained in one day by the analysis of four samples independently sampled but prepared identically. In turn, to estimate the intermediate precision of injection and sampling, separations within 3 days were carried out, respectively, four repetitions from the same sample and 4 independently prepared samples. In the next stage, the intermediate precision in relation to the capillary was also verified. For this, four samples with the use of two different capillaries were subjected to a single analysis.

The results calculated as the standard deviation and relative standard deviation (RSD) expressed in % (the coefficient of variation, CV) for the relative migration times of the peaks (the migration times for the analyte divided by the time of EOF signal) are listed in Table 2.

Considering injection and sampling repeatability, CV values were opposite to those assumed. CV for injection is greater than for repeatability of the sampling. The same situation is in the case of the intermediate precision—the CV values for sampling were better than for the intermediate precision of injection. Based on the above results, it was found that the most probable cause is multiple capillary insertions into a single PCR tube. This may be due to the large temperature amplitude between the sample storage module (maintained at T = 10 °C) and the separation temperature (35 °C). The tip of the capillary, immersed in the sample in a PCR tube, may cause changes in the chemical composition of the sample matrix (e.g., speed up the evaporation of the most volatile components or changes related to diffusion or temperature transfer). Therefore, to improve the injection precision, it was decided to verify lower temperature conditions in the sample storage module (10, 15, and 20 °C) so that the above-mentioned amplitude was smaller. Comparing the obtained injection repeatability, it was found that similarly satisfactory results were obtained for the temperatures of 15 °C and 20 °C, namely CV = 0.27 and 0.26, respectively. However, better separation of the peaks was observed at 15 °C; thus, it was selected for further stages of the research.

Taking into account the CV values corresponding to the intermediate precision compared to the repeatability of the method, one should remember that, whenever possible, a comparative analysis of unknown and reference samples should be performed in one day using the same capillary.

### 2.5. Analysis of Real Samples

The next step was to analyze a collection of real samples—polyester fabrics secured from various clothes. The collected samples varied considerably in terms of countries of origin, distributor, products from which they were taken (clothing or decorative elements of houses), the chemical composition of the fabric, and color (details in Section 3.2 in Table 7). All 28 samples were divided into four groups considering their color and then treated according to the presented above procedure (extraction and separation of the extract components with the use of the developed MEEKC method). 

It is worth emphasizing, that the partial differentiation of the samples was possible as early as at the stage of extraction, due to the diversity of colors of obtained extracts. This observation is most valuable in the case of fabrics that did not differ in color before extraction. This phenomenon can simply be explained by the solvent (extractant) interaction with the chromophores of dissolved molecules of one or more dyes (mixtures of several dyes are usually used to dye the fibers) and cause shifting in the maximum radiation absorption in the visible range. As a result, extracts obtained, for example, from black-colored fabrics, can have a color that falls into green or blue shades. This effect is not spectacular, but it is sufficiently noticeable that it can be used in the initial stage of differentiation.

A database was created, including the following obtained results:Total number of signals present at the electropherograms registered at 220 nm (signals with intensity above 0.5 mAU);Number of dye signals selected based on the UV-Vis spectra (including dye signals with intensity above 0.4 mAU at 220 nm);Bands in the Vis absorption spectrum recorded for dye signals—information corresponding to the color of analyzed dyes;Relative migration times of analytes, calculated in relation to the migration time of electroosmotic flow (EOF) signal, treated similarly to an internal standard.

To facilitate the comparison of the relative migration times of the analytes, a simple scheme was developed basing on the results obtained in the method validation process. Measurements for real samples were carried out over at least a few days, therefore the results (SD) obtained during the intermediate precision of sampling were used. The expanded uncertainty (*U, k =* 2) of the relative migration times of 9 individual peaks was determined. The values of this parameter were in the range 0.02–0.11. It is worth emphasizing that in CE, along with the increase in the migration times of analytes, a deterioration in the repeatability is usually observed due to the time-dependent changes in EOF (caused, among others, by the modification of the electrical double layer, which in turn is a consequence of the adsorption of molecules to silanol groups on the inner wall of the capillary or by the deepening dispersion of the analyte). However, the use of relative migration times allows for reducing this effect. Furthermore, the CV values obtained do not directly reflect the described phenomenon (signal 6 is characterized by much lower repeatability than signals 7, 8, and 9 occurring at longer migration times). Therefore, it can be concluded that the repeatability of the results depends largely on the type of analyte tested and the way it interacts with ME during measurement and less on the EOF stability. However, in forensic comparative studies, the analysts do not have information on specific analytes present in the sample and do not have the validation parameters assigned to them that reflect repeatability. For this reason, it is necessary to determine the limits of expanded uncertainty values that would allow the assessment of the quality of the similarity between the relative migration times of the analytes of compared samples. Using the maximum expanded uncertainty value obtained in method validation, two groups were created to assess the similarity of the results obtained (details in Table 3).

### 2.6. Application of the Developed Method for the Comparison Study of Polyester Fibers

To confirm the possibility of using a developed separation method for the differentiation of fiber extracts, the intralaboratory test was carried out. The first stage of the test had to reflect the situation of the comparative study of the fibers. To avoid the analyst’s self-suggestion, new symbols were assigned to the samples taken for the test. For this purpose, three black fabrics B1–B3 (comparative material) were selected from the collection of twenty-eight available fabrics, and thread X (evidence material) was taken from one of them. The analyst’s task was to determine from which of the three comparative materials B1–B3 the secured thread X originated. The factors used for differentiation steps were as follows: extract color, the total number of signals, the number of dye signals, absorption bands in the Vis spectrum, and relative migration times of analytes. 

As mentioned in Section 3.5. the differences in the extract colors were possible to be visually detected. The colors of the extracts from samples B2 and X showed a high similarity (a noticeable characteristic green shade), which was the first indication of the similarity of the samples. A MEEKC analysis was performed to confirm the preliminary results. The data obtained, including the number of signals (total and dye), the Vis radiation absorption ranges (corresponding to the dye colors), and the calculated relative migration times of the analytes are summarized in Table 4.

The ranges of Vis radiation absorption were detected using photodiode array detection (PDA). The fragments of PDA electropherogram views showing the dye signals are presented in Figure 4 and the electropherograms at 220 nm are in Figure 5.

The total number of signals, as so the number of dye signals, make it possible to exclude the common origin of samples B3 and X. The number of dye signals and the absorption bands in the Vis spectrum showed the differences in the composition of extracts from samples B1 and X. The same number of signals (total and dye), and the absorption bands in the Vis spectrum (dye color) indicate the similarity of samples B2 and X. The calculated *U* values are lower than 0.11 for all peaks and show very good similarity. The results led to the conclusion that samples B2 and X were, with very likelihood, preserved from the same material.

### 2.7. Application of the Developed Method for the Identification Study of Polyester Fibers

The next step of the intralaboratory test was made to reflect the identification study of fibers. For this purpose, all 28 real samples were divided into five groups, based on color: red (1), green (2), blue (3), black (4), and other (5) (details shown in Figure 6).

A single fabric from each created group (1–4) was picked up, and then the thread samples were collected (named, respectively, A1, A2, A3, and A4; new symbols were assigned to avoid analyst self-suggestion) and analyzed according to the developed procedure. Then, the information obtained from electropherograms, and PDA views was compared to the database, within four coloristic groups. An example of this comparison is presented in Figure 7, for the identification of the extracts from the group of green textiles (2).

Already by analyzing the electrophoretic profiles at 220 nm, significant differences between the samples can be noticed. What is worth emphasizing, the presented electropherograms clearly show the shifts of signals generated by EOF (especially for sample P16a compared to the other samples). This creates the need to use more reliable methods than just visual analysis of obtained electropherograms, for comparing the data for individual samples. Therefore, the relative migration times of the analytes were calculated and compared as in Section 2.5. The details are shown in Table 5.

Similarly, as in the first stage of the test, the numbers of signals (total and dye) exclude the common origin of samples A2 with P2, P8, and P25. The Vis spectra indicate the differences in the chemical composition of samples P21 and A2. The total number of signals, as well as the number of dyes (establish according to the range of Vis radiation absorption corresponding to the color of dyes used in dying mixtures), give a basis for considering the common origin of samples A2 and P16a. The calculated *U* values for signals presented in electropherograms for both compared samples are lower than 0.11 which shows a very good similarity of the samples (see Table 5). It can be concluded that samples A2 and P16a come from the same source.

The same identification procedure was performed within groups A1, A3, and A4, details presented in Appendix A. In group A3 the identification of samples was possible already by comparison of the total number of signals, in A4 by the number of dye signals, and in A1 by absorption bands in the Vis spectrum. In each case, the *U* values for the compared samples were lower than 0.11, which makes it possible to consider samples A1 with P26, A3 with P9, and A4 with P20 as similar.

## 3. Materials and Methods

### 3.1. Materials

Reagents used during the experiment, such as chlorobenzene, sodium tetraborate decahydrate, potassium dihydrogen phosphate, phosphoric acid, sodium dodecyl sulfate, sodium docusate, sodium cholate, Brij-35, Tween 20, 2-hydroxypropyl-β-cyclodextrin (average molecular weight 1.540) and 2-hydroxypropyl-γ-cyclodextrin (0.6 molar substitution) were purchased from Sigma Aldrich (Darmstadt, Germany). Sodium hydroxide and hydrochloric acid were supplied by POCH (Gliwice, Poland), whereas n-heptane by Chempur (Piekary Śląskie, Poland), isopropanol and methanol by Honeywell (Offenbach, Germany), and butan-1-ol, n-octane and DMSO by MERCK (Darmstadt, Germany). The water solutions were made using ultrapure water generated in-lab by the MiliQ MERCK-Millipore system (Darmstadt, Germany). 

### 3.2. Samples

Nine commercially available dyeing mixtures (D1–D9, Table 6) were used in the optimization of the separation conditions process. These powder mixtures were composed of disperse dyes (single or multiple) and other colorless excipients. The dyes contained in the analyzed mixtures along with the commercial names and the names according to the Color Index (CI) [47] (C.I. Generic Name (CIGN) related to the usage class of dye and C.I. Constitution Number (CICN) related to the structure of dye) are listed in Table 6. 

To optimize extraction conditions three polyester fabrics made of polyethylene terephthalate (PET), originating from Miranda Textiles (Turek, Poland) were used: orange (FA), light green (FB), and light grey (FC). They were colored with ternary mixtures of disperse dyes in the laboratory of the Faculty of Material Technologies and Textile Design of the Lodz University of Technology [46]. In the course of the research, a colorless fabric (FD), ready for dyeing, was also used. Details are presented in Table 6, and photos of fabrics are provided in Figure 8. 

The real samples (P1–P26) were collected from clothing. Most of them were fabrics made entirely of polyester or containing a predominance of polyester in the composition. A detailed description of the real samples tested is included in Table 7.

### 3.3. Instrumentation

In this study, a PA 800 Plus capillary electrophoresis system with a PDA detector (190–600 nm) and Karat 32 software version 9.1 (Beckman-Coulter, Brea, CA, USA) was used. Separation was carried out for 30 min in a polyimide-coated fused silica capillary of a total length of 50 cm (effective length of 40 cm to the detector window) and an internal diameter of 50 μm. The separation voltage was 30 kV, and the separation temperature was 25, 30, or 35 °C, depending on the experiment. The sample was injected hydrodynamically at 0.7 psi for 6 s. Between injections, samples were stored at 15 °C in the sample storage to minimize evaporation of volatile components of BGE. Before each measurement sequence, the capillary was conditioned by rinsing for 5 min (pressure 20 psi) with methanol, 5 min with 1 M HCl, 1 min with water, 5 min with 0.1 M NaOH, 1 min with water, and 10 min with background electrolyte with an applied voltage of 20 kV. Between measurements, the capillary was rinsed for 4 min with methanol, 1 min with 1M HCl, 30 s with water, 5 min with 0.1 M NaOH, and 2.5 min with BGE. Immediately before insertion into the capillary, all solutions were degassed by centrifugation at 10,000× *g* rpm (Allegra X-35 Centrifuge, Beckman Coulter, Brea, CA, USA). All necessary statistical calculations were performed using the Statistica 13.3 PL software (TIBCO Software, Palo Alto, CA, USA).

### 3.4. Stock ME and BGE Preparation Procedure

The stock ME was prepared by weighing the organic phase (n-octane and butan-1-ol), directly in the beaker. Then, solid surfactants were added to the beaker, carefully to prevent surfactants from depositing on the walls of the beaker. The aqueous phase—40 mM borate buffer—was then dropped into the beaker at a rate of about 2 drops per second. Then, the mixture was sealed with parafilm and placed in an ultrasonic bath (Polsonic, Warsaw, Poland) for 30 min and set aside overnight at ambient temperature.

In the next step, the stock ME was mixed with isopropanol in the appropriate volume ratio, depending on the research stage. Finally, solutions of cyclodextrins: 2-hydroxypropyl-β-cyclodextrin and 2-hydroxypropyl-γ-cyclodextrin (separately or mixed in a 1:1 molar ratio, concentrations depending on the experiment) in stock ME with the addition of isopropanol were prepared. The BGE prepared in this way was then filtered using a nylon syringe filter with a pore diameter of 0.45 µm (Labex Ltd., Budapest, Hungary). 

### 3.5. Sample Preparation Procedures

For optimization and validation of the developed method, methanol stock solutions of commercially available dyeing mixtures (D1–D9, Table 6) at a concentration of 1 mg·mL^−1^ were used as standards. In total, 10 μL of each of the stock solutions were placed in a PCR tube, then the solvent was evaporated under a nitrogen atmosphere at 40 °C using a sample concentrator (Liebisch, Germany). Subsequently, the residues were dissolved in 50 µL of BGE, centrifuged for 5 min at 12,000× *g* rpm (Microfuge 16 Centrifuge, Beckman Coulter, Krefeld, Germany), and analyzed.

Polyester threads of 1 or 4 cm length (consisting of approximately 50 single fibers) were used as a sample. Extraction was carried out in glass vials closed with caps with 50 μL of chlorobenzene at 100 °C in a thermoshaker (VWR, Radnor, PA, USA) operating at 1000 rpm for 0.5, 1, or 6 h, depending on the experiment. The preparation of five samples (P10, P13a, P13b, P17, P18) required some adjustments (see Table 8 for details).

The extracts were transferred to PCR tubes with the use of a micro syringe to avoid transferring microfibers. The extracts were then evaporated to dryness under gentle nitrogen flow (40 °C) and dissolved similarly to dye standards.

## 4. Conclusions

The developed MEEKC method is characterized by a high analytical potential even in the case of such difficult samples as dyeing mixtures of non-ionic disperse dyes with various chemical structures. Based on the conducted research, it can be concluded that it is a useful tool mainly because of the multitude of parameters that can be modified: instrumental parameters of the separation process and the composition of ME constituting the background electrolyte, and thus the interactions occurring in the system during the separation process. The introduction of cyclodextrins to BGE significantly improved the resolution and shortened the analysis time. The best results were obtained for microemulsions with a mixture of two cyclodextrins.

Carrying out the Doehlert matrix design enables the determination of the optimal conditions for MEEKC separation in only 13 experiments. The best BGE consisted of base ME (1.5% (*w*/*w*) sodium cholate, 1.5% (*w*/*w*) SDS, 0.8% (*w*/*w*) n-octane, 6% (*w*/*w*) butan-1-ol, 90.2% (*w*/*w*) 10 mM sodium tetraborate) modified by 30% (*v*/*v*) iPr and the addition of two CDs (20 mM 2-hydroxypropyl-β-cyclodextrin and 20 mM 2-hydroxypropyl-γ-cyclodextrin). The separation process was maintained at 35 °C and a high voltage of 30 kV was applied. The optimal conditions of the dye extraction process turned out to be as follows: thread length 1 cm, extraction time 6h, T = 100 °C, rotation speed 1000 rpm, extractant: chlorobenzene. The MEEKC method optimized in this study was successfully applied to the analysis of 26 polyester fabrics with acceptable repeatability (relRSD_tm_ ≤ 0.38%) and intermediate precision (relRSD_tm_ ≤ 3.21%). A qualitative approach was sufficient to conduct a successful comparative examination of polyester fibers.

Certainly, further research is needed to improve the developed method. One direction is to adapt the method to analyze and differentiate fibers similar in size to those that are secured in practice at the scene of a crime. However, it should be noted that to the authors’ knowledge, no method using capillary electrophoresis has yet been developed for the analysis of extracts from polyester fibers of such small sizes [33].

For this reason, the authors consider increasing the sensitivity of the method by optimizing the on-line stacking process, by using laser induced fluorescence (LIF) or mass spectrometer (MS) detector. In the case of the latter, another advantage is the additional information obtained in the form of MS spectra, which gives even greater possibilities of identification. Unfortunately, the surfactants used in the ME of the proposed method are incompatible with the mass spectrometer as they are non-volatile and may contaminate the ion source. Potential solutions include the development of a new background electrolyte for the nonaqueous capillary electrophoresis (NACE) technique, in which the obtained resolution, combined with the possibility of extracting information about a specific analyte based on mass spectra, may prove sufficient to conduct reliable analyses. In another case, the solution may be to use ME containing volatile surfactants or ME without surfactants [48].

## Figures and Tables

**Figure 1 molecules-27-06974-f001:**
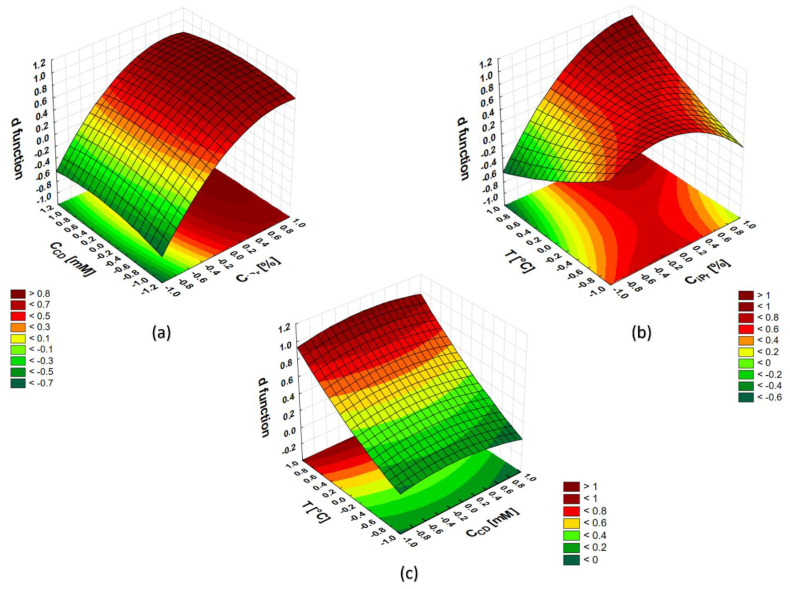
Response surfaces *d* vs. c_iPr_ and c_CD_ (**a**), *d* vs. c_iPr_ and T (**b**), *d* vs. c_CD_ and T (**c**): where c_iPr_ is the concentration of isopropanol (% *v*/*v*) and c_CD_ is the total concentration of cyclodextrins HP-β-CD and HP-γ-CD mixed in 1:1 molar ratio), and T is the separation temperature.

**Figure 2 molecules-27-06974-f002:**
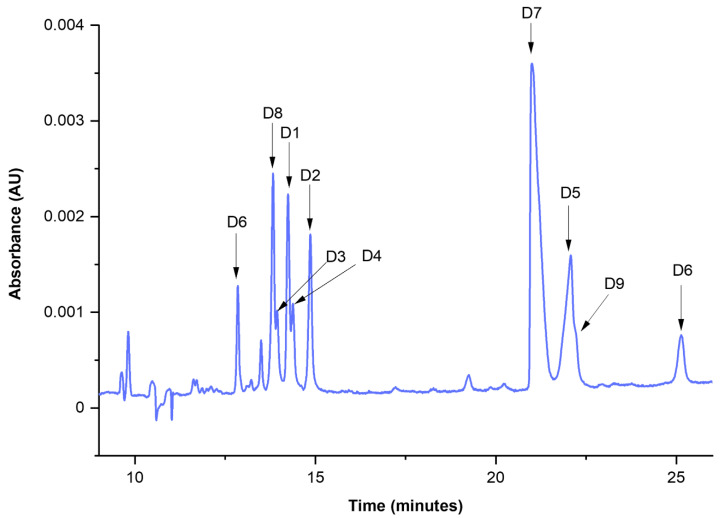
Microemulsion electrokinetic capillary chromatography (MEEKC) separation of 9 dye standards (D1–D9) accomplished in optimal conditions (analytical wavelength—220 nm, background electrolyte (BGE)—ME8.1).

**Figure 3 molecules-27-06974-f003:**
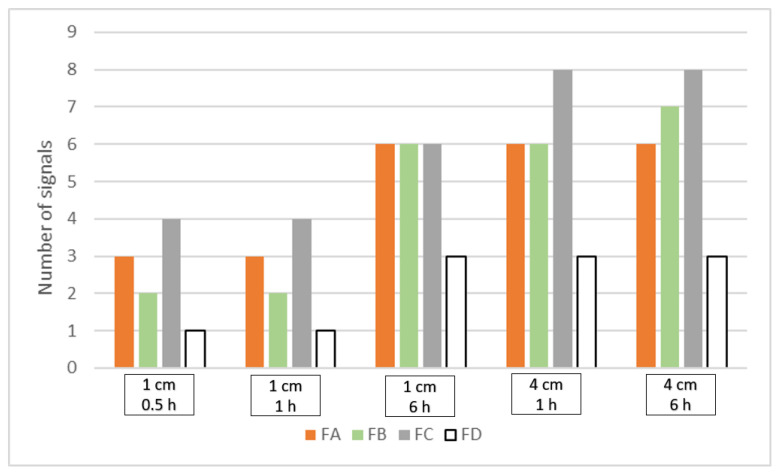
Comparison of a number of signals detected on the electropherograms registered in 220 nm for the extracts obtained in different high temperature extraction (HTE) conditions from four standard fabrics FA-FD.

**Figure 4 molecules-27-06974-f004:**
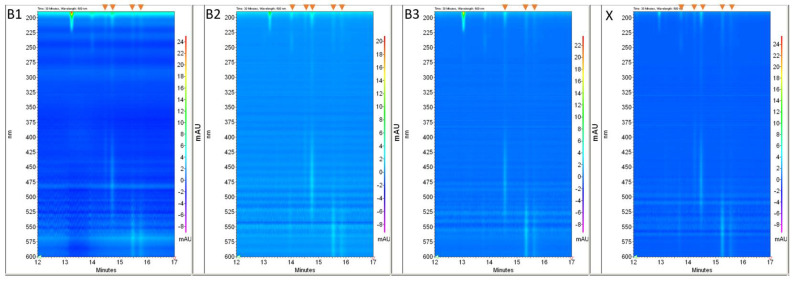
Fragments of PDA views registered for samples B1–B3, and X. Dye signals are marked on the top, by the orange arrows.

**Figure 5 molecules-27-06974-f005:**
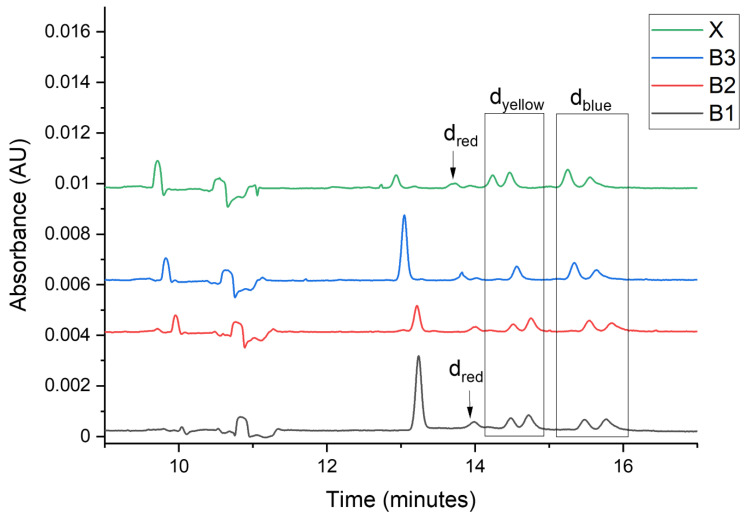
Electropherograms registered for samples B1–B3, and X at 220 nm, d_red_—red dyes, d_yellow_—yellow dyes, d_blue_—blue dyes.

**Figure 6 molecules-27-06974-f006:**
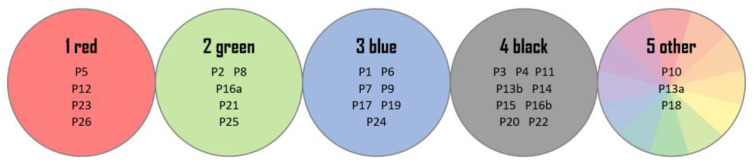
The color groups of fabrics and symbols of the samples belonging to them.

**Figure 7 molecules-27-06974-f007:**
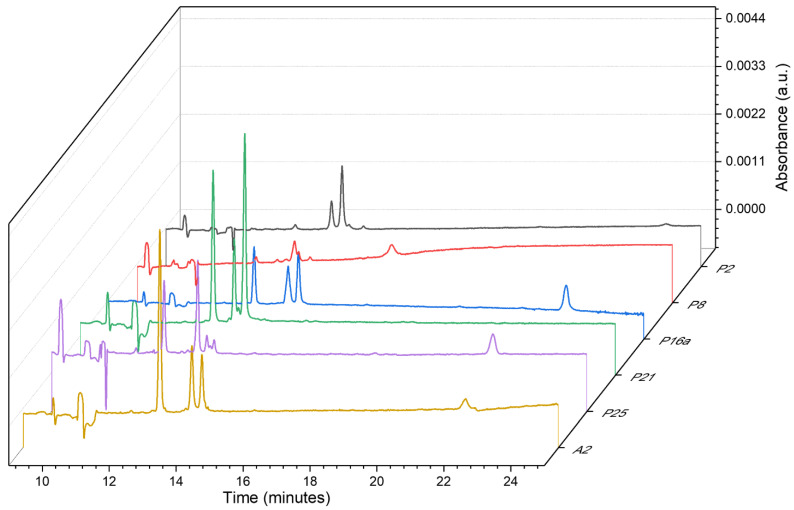
Comparison of electropherograms with dye signals, recorded at 220 nm for the extracts from fibers included in group 2 from green fabrics and sample A2, secured from one of the textiles.

**Figure 8 molecules-27-06974-f008:**
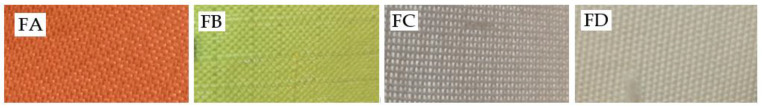
Fabrics used in experiments FA-orange, FB-light green, FC-light grey, and FD-colorless.

**Table 1 molecules-27-06974-t001:** Doehlert design with coded and real experimental values of variables and measured responses; c**_iPr_** is the concentration of isopropanol and c**_CD_** is the total concentration of 2-hydroxypropyl-β-cyclodextrin (HP-β-CD) and 2-hydroxypropyl-γ-cyclodextrin (HP-γ-CD) mixed in a molar ratio 1:1, T is separation temperature, *a* is the parameter with values 1 or 0, *F* is the response function.

No.	Coded Variables			Real Variables			Measured Response			
	x_1_	x_2_	x_3_	c_iPr_ [% *v*/*v*]	c_CD_ [mM]	T [°C]	*n*	*t_D_* _1_	*a*	*F*
1	0	0	0	15	40	30	7	11.74	1	0.60
2	0	1	0	15	60	30	6	11.98	1	0.50
3	0.866	0.5	0	30	50	30	8	18.42	1	0.43
4	0.289	0.5	0.817	20	50	35	9	10.43	1	0.86
5	0	−1	0	15	20	30	6	11.79	1	0.51
6	−0.866	−0.5	0	0	30	30	7	9.67	1	0.72
7	−0.289	−0.5	−0.817	10	30	25	7	12.10	1	0.58
8	−0.866	0.5	0	0	50	30	7	7.85	0	0.00
9	−0.289	0.5	−0.817	10	50	25	8	12.55	1	0.64
10	0.866	−0.5	0	30	30	30	8	16.78	1	0.48
11	0.577	0	−0.817	25	40	25	9	18.90	1	0.48
12	0.289	−0.5	0.817	20	30	35	8	10.33	0	0.00
13	−0.577	0	0.817	5	40	35	7	7.14	0	0.00

**Table 2 molecules-27-06974-t002:** The results of validation—average relative migration times, standard deviation (SD), and coefficient of variation (CV) of 9 peaks corresponding to standard disperse dyes.

Peak	1 (D8)	2 (D3)	3 (D1)	4 (D4)	5 (D2)	6 (D7)	7 (D5)	8 (D9)	9 (D6)
**Injection repeatability (1 sample × 4 repetitions × 1 day = 4 runs), 3-day average**									
rel t_m, av_ [min]	1.37	1.39	1.41	1.43	1.46	2.05	2.11	2.13	2.37
SD	0.01	0.01	0.01	0.02	0.01	0.01	0.01	0.01	0.01
CV [%]	0.72	0.91	0.82	1.24	0.86	0.42	0.43	0.39	0.32
**Sampling repeatability (4 samples × 1 repetition × 1 day = 4 runs), 3-day average**									
rel t_m, av_ [min]	1.39	1.40	1.43	1.44	1.47	2.01	2.09	2.10	2.34
SD	0.01	0.01	0.01	0.01	0.01	0.01	0.003	0.002	0.004
CV [%]	0.60	0.63	0.71	0.52	0.65	0.56	0.16	0.11	0.19
**The intermediate precision of injection (1 sample × 4 repetitions × 3 days = 12 runs)**									
rel t_m, av_ [min]	1.37	1.39	1.41	1.43	1.46	2.04	2.11	2.12	2.37
SD	0.01	0.02	0.01	0.02	0.01	0.07	0.05	0.05	0.06
CV [%]	0.58	0.72	0.62	1.03	0.78	3.21	1.67	1.61	2.13
**The intermediate precision of sampling (4 samples × 1 repetition × 3 days = 12 runs)**									
rel t_m, av_ [min]	1.39	1.40	1.42	1.44	1.47	2.01	2.09	2.11	2.34
SD	0.01	0.02	0.01	0.01	0.01	0.05	0.02	0.02	0.03
CV [%]	0.88	1.08	0.99	0.93	1.01	2.67	0.87	0.88	1.28
**The intermediate precision in relation to the capillary (4 samples × 1 repetition × 2 capillaries × 1 day = 8 runs)**									
rel t_m, av_ [min]	1.39	1.41	1.43	1.45	1.47	2.03	2.09	2.11	2.35
SD	0.01	0.01	0.02	0.01	0.02	0.06	0.02	0.02	0.04
CV [%]	1.02	1.04	1.15	0.90	1.21	2.93	1.05	1.03	1.50

**Table 3 molecules-27-06974-t003:** Limits of CV and range values defining the similarity of samples in the comparative examination.

Similarity	Similar	Different
**expanded uncertainty (*U, k* = 2)**	≤0.11	>0.11
**annotations**	*U* values lower than parameters obtained in method validation	*U* values higher than parameters obtained in method validation

**Table 4 molecules-27-06974-t004:** Results of differentiation of samples B1–B3 and X; yellow, red, and blue colors corresponding to signals with the maximum absorption bands at 380–490, 490–580, and >580 nm, respectively.

Signal Number	Relative Migration Times of the Analytes				*U* (*k =* 2) for Samples B2 and X
	B1	B2	B3	X	
1	1.32	1.32	1.33	1.33	0.02
2	1.39	1.40	1.41	1.41	0.02
3	1.44	1.44	1.48	1.47	0.03
4	1.47	1.47	1.56	1.49	0.03
5	1.54	1.54	1.59	1.57	0.04
6	1.57	1.57	-	1.60	0.05
**total**	6	6	5	6	
**dye**	4	5	3	5	

**Table 5 molecules-27-06974-t005:** Results of differentiation of samples from a group of green fabrics (2) and sample A2; yellow, red, and blue colors corresponding to signals with the maximum absorption bands at 380–490, 490–580, and >580 nm, respectively.

Signal Number	Relative Migration Times of the Analytes						*U* (*k =* 2) for Samples P16a and A2
	P2	P8	P16a	P21	P25	A2	
1	1.46	1.48	1.33	1.32	1.24	1.32	0.01
2	1.49	1.79	1.43	1.38	1.33	1.42	0.02
3	-	-	1.46	1.40	1.44	1.45	0.02
4			2.26	1.42	1.47	2.25	0.01
5			-	-	1.49	-	-
6					2.39		
**total**	2	2	4	4	6	4	-
**dye peaks**	2	2	2	2	4	2	

**Table 6 molecules-27-06974-t006:** Commercially available dyeing mixtures used in experiments.

Mixture Symbol	Commercial Name	Dyeing Component CIGN, CICN	Color of Methanolic Solution	Fabric Symbol (Color)
D1	Setapers scarlet SE-3GFL	Disperse Red 54 (11131)	red	FA (orange)
D2	Serilene blau HBGL 200%	Disperse Blue 73 (63265)	blue	
D3	Serilene pink A-2GN	Disperse Red 86 (62175)	rose	
D4	Serilene blau RG 200%	Disperse Blue 165 (11077) Disperse Blue 60 (61104)	blue	FB (light green)
D5	Serilene yellow 6 GLS 200%	Disperse Yellow 114 (128455)	yellow	
D6	Serilene golden yellow K-2 G	Disperse Orange 25 (11227) Disperse Yellow 86 (10353) Disperse Yellow 211 (12755)	yellow-orange	
D7	Serilene yellow A-GL	Disperse Yellow 42 (10338)	yellow	FC (light grey)
D8	Serilene red A-TB	Disperse Red 92 (60752)	red	
D9	Serilene blau A-BL	Disperse Blue 77 (60766)	blue	

**Table 7 molecules-27-06974-t007:** The real sample used in experiments.

Sample	Fabric Composition	Color	Distributor (Origin Country)	Clothing Type
P1	100% polyester	navy	OTCF S.A. (Poland)	sweatshirt
P2	-	green	MiLady (Italy)	dress
P3	50% polyester 50% viscose	black	unknown (unknown)	sweater
P4	50% cotton 50% polyester	grey	Medicine (China)	sweatshirt
P5	100% polyester	red	Asos (China)	dress lining
P6	100% polyester	navy	Reserved (China)	jacket
P7	100% polyester	blue	Mohito (China)	sweater
P8	100% polyester	green	Asos (China)	dress
P9	100% polyester	blue	Janina (China)	dress
P10	100% polyester	gold	MANGO (Spain)	skirt lining
P11	100% polyester	black	MISSGUIDED (UK)	top
P12	100% polyester	red	Pepco (unknown)	pillowcase
P13a *	95% polyester 5% spandex	rose	Mohito (Poland)	dress
P13b *	unknown	black	Mohito (Poland)	dress
P14	94% polyester 6% spandex	black	AMISU (China)	skirt
P15	90% polyester 10% spandex	black	F&F (India)	top
P16a	85% polyester 13%viscose 2% spandex	green	House (China)	skirt
P16b	85% polyester 13%viscose 2% spandex	black	House (China)	skirt
P17	64% polyamide 28% polyester 8% spandex	blue	Primark (China)	swimsuit
P18	73% cotton 23% polyester 4% spandex	black	H&M (China)	shirt
P19	100% polyester	navy	Alena firany (Poland)	curtains
P20	100% polyester	black	Sinsay (China)	top
P21	100% polyester	green	Reserved (China)	dress
P22	95% polyester 5% spandex	black	Mohito (Cambodia)	jumpsuit
P23	unknown	red	Labres LINGERIE (Poland)	nightshirt
P24	100% polyester	blue	Eva & Lola China	dress lining
P25	94% polyester 5% spandex	green	Sinsay (Burma)	dress
P26	unknown	red	unknown (unknown)	hairband

* a or b—samples taken from different parts of the same garment.

**Table 8 molecules-27-06974-t008:** Modifications of extraction procedure for samples P10, P13a, P13b, P17, P18.

Sample Symbol	Comment	Adjustment
P10 P17	Poor extraction yield: only a pale beige color was observed after reducing the solvent volume	The amount of sample increased to a sheet of fabric ca. 0.5 × 0.5 cm
P13a P13b	Poor extraction yield: colorless extract. The fibers were not entirely colored, only the top layer of the fabric was colored,	The amount of sample increased to a sheet of fabric ca. 0.5 × 0.5 cm
P18	Poor extraction yield: colorless extract. Probably due to the high content of cotton and low content of polyester in the fabrics which indicate dying with other classes of dyes than disperse ones	Excluded from the study

## Data Availability

Not applicable.

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
