# Peer review of "Development of the Microemulsion Electrokinetic Capillary Chromatography Method for the Analysis of Disperse Dyes Extracted from Polyester Fibers"

_molecules, 2022, doi:10.3390/molecules27206974_

Round 1

Reviewer 1 Report

Dear Editor, I hope you are doing well

I have revised the manuscript, the authors need to check the language of the manuscript, and the methods section should be enhanced also the results and discussion.

Here are the comments:-

Abstract needs to be reformulated regarding adding relevant results, and logistical and structural errors

Rephrase lines 53-84 for clarity

Clear the study objectives at the end of the introduction

Line 151, and 157 adjust the equation

Check language by expert

Check and update the outputs of all references

Check the name of all devices, chemicals, and their companies

Enhance the resolution of figure 4, and 6

Reformulate the head of Table 8

Check the abbreviations in the whole manuscript

Reviewer 2 Report

This article is very well written. From the design of the experiment to the implementation, from the theory to the data to the analysis, it is impeccable. I don't think we need to make any changes to publish it.

Reviewer 3 Report

Dear Authors

Please see comments in attached review!

kind regards

Reviewer

Round 2

Reviewer 1 Report

The authors have carefully processed all comments. The quality of the manuscript has increased significantly. I have no further comments.